# Aquaculture Production and Its Environmental Sustainability in Thailand: Challenges and Potential Solutions

**Tiptiwa Sampantamit [1,2,\*], Long Ho [1], Carl Lachat [3], Nantida Sutummawong [2], Patrick Sorgeloos [4] and Peter Goethals [1]**

[1] Department of Animal Sciences and Aquatic Ecology, Faculty of Bioscience Engineering, Ghent University, 9000 Ghent, Belgium; Long.TuanHo@UGent.be (L.H.); Peter.Goethals@UGent.be (P.G.)

[2] Department of Biological and Environmental Sciences, Faculty of Science, Thaksin University, 93110 Patthalung, Thailand; sunantida@tsu.ac.th

[3] Department of Food Technology, Safety and Health, Ghent University, 9000 Ghent, Belgium; Carl.Lachat@UGent.be

[4] Laboratory of Aquaculture and Artemia Reference Center, Faculty of Bioscience Engineering, Ghent University, 9000 Ghent, Belgium; Patrick.Sorgeloos@UGent.be

\* Correspondence: Tiptiwa.Sampantamit@ugent.be; Tel.: +32-484175881

**Abstract:** Though aquaculture plays an important role in providing foods and healthy diets, there are concerns regarding the environmental sustainability of prevailing practices. This study examines the trends and changes in fisheries originating from aquaculture production in Thailand and provides insights into such production's environmental impacts and sustainability. Together with an extensive literature review, we investigated a time series of Thai aquaculture production data from 1995 to 2015. Overall, Thai aquaculture production has significantly increased during the last few decades and significantly contributed to socio-economic development. Estimates of total aquaculture production in Thailand have gradually grown from around 0.6 to 0.9 million tons over the last twenty years. Farmed shrimp is the main animal aquatic product, accounting for an estimated 40% of total yields of aquaculture production, closely followed by fish (38%) and mollusk (22%). Estimates over the past decades indicate that around 199470 ha of land is used for aquaculture farming. Out of the total area, 61% is used for freshwater farms, and 39% is used for coastal farms. However, this industry has contributed to environmental degradation, such as habitat destruction, water pollution, and ecological effects. Effective management strategies are urgently needed to minimize the environmental impacts of aquaculture and to ensure it maximally contributes to planetary health. Innovative and practical solutions that rely on diverse technology inputs and smart market-based management approaches that are designed for environmentally friendly aquaculture farming can be the basis for viable long-term solutions for the future.

**Keywords:** aquaculture production; sustainability; environment; Thailand

---

## 1. Introduction

As stated in the Sustainable Development Goals (SDGs), there is a global concern about erasing malnutrition, improving poverty alleviation, and achieving food security and planetary health. In particular, SDGs 1 and 8 are related to poverty and economic growth, respectively, and SDGs 2, 3, and 12 are about zero hunger, good health, and responsible consumption and production, respectively [1]. The importance of fisheries as a source of food and nutrition cannot be overstated, especially in the face of population growth and increasing demand for animal protein [2,3].

Several studies have indicated that fish is an excellent source of animal proteins, micronutrients, and vitamins [4–7].

Globally, fisheries production peaked at about 171 million tons in 2016, of which aquaculture production represented 80 million tons (47%) and capture production represented 91 million tons (53%) [8]. During the recent decades, a large number of the world's fish stocks have been depleted, and, therefore, global fisheries are no longer capable of producing their maximum sustainable yield [9]. Aquaculture has contributed to the impressive growth in the seafood supply for human consumption [10]. Thailand's aquacultural sector has rapidly developed during the last few decades and has been accompanied with tangible socio-economic development. The country was ranked among the top twenty-five countries in terms of fisheries production in 2018 [8]. Recent statistics that were collected by the Department of Fisheries (DoF) [11] estimate that Thailand's aquaculture production in 2016 exceeded more than 0.9 million tons, of which 0.5 million tons (57%) were from coastal aquaculture and 0.4 million tons (43%) were from freshwater aquaculture.

The growing production of freshwater and marine aquaculture has tremendous potential to help sustainably feed the growing human population [12]. However, several studies have pointed towards the harmful effects of aquaculture production and, in particular, its environmental and ecological impacts. For example, the rapid growth in shrimp farming is a key driver of mangrove forest degradation and reduction of natural habitats and biodiversity [13–18]. Additionally, aquaculture production may lead to a decrease in biodiversity and nutrition diversity, as it usually focuses on a few selected species [3,19–21].

This study examines the trends and changes in fisheries originating from aquaculture production in Thailand and provides insights into such production's environmental impacts and sustainability. First, we describe aquaculture production in Thailand, including the volume and value of aquaculture production and the diversity of farmed species. Second, we review the contribution of the development of aquaculture production to environmental degradation in Thailand. Finally, the possible measures that are needed to reach a sustainable future for Thai aquaculture production are presented. Our analysis focuses on Thailand's aquaculture production data by using a time series of DoF statistics from 1995 to 2015.

## 2. Data Collection and Methods

Data on aquaculture production were obtained from the fisheries statistical yearbooks that have been published by the DoF while using a time series from 1995 to 2015. Aquaculture is the culture of aquatic organisms, which includes fish, mollusk, and crustaceans. The aquaculture production yield is reported as weights of fresh products in Table 1.

We used the methodology of Nesbitt, et al. [22] as a reference to identify the common name, scientific name, genus, and family of fishes and shellfishes. All species that were mentioned in the database of the DoF were identified based on a guidebook of marine fishes in Thailand, the global fish database Fishbase (http://www.fishbase.org), the International Union for Conservation of Nature and Natural Resources (IUCN) Red List of Threatened Species (http://www.iucnredlist.org/about), and Species 2000 and the Integrated Taxonomic Information System (ITIS) Catalogue of Life, (www.catalogueoflife.org/col). Details of these databases can be found in Table A1 in Appendix A.

The total yield of each species and group were calculated based on their annual yields. Then, we calculated the relative abundance of the species that were produced in Thai aquaculture from 1995 to 2015 as the percentage of their weight. We focused on major species that are important in aquaculture; see Figure 1. The maps in Figure 2 that display land changes were created by Quantum Geographic Information System (QGIS) version 3.2.2.

Peer-reviewed studies on aquaculture in Thailand, written in both Thai and English, were used as reference and discussion points. This study also used several official reports, such as the master plan on Thailand's aquaculture development [23] and the National Economic and Social Development Plan [24].

### 3. Trends in Aquaculture Supply in Thailand

*3.1. Yield of Aquaculture Production*

Aquaculture production in Thailand is broadly divided into two categories: (1) inland freshwater aquaculture and (2) coastal or marine aquaculture [25]. Table 1 illustrates Thailand's aquaculture production between 1995 and 2015. Over the last twenty years, on average, the annual aquaculture production was about one million tons per year (range of 500000–1400000 tons). Aquaculture production yield increased from around 553600 tons in 1995 to 928500 tons in 2015 [26–46]. About 62% (617900 tons) of the annual production yield was from coastal aquaculture, while the other 38% (384600 tons) was from freshwater aquaculture.

Based on the available database of the DoF, the three main aquaculture products were shrimp, fish, and mollusk. Farmed shrimp was the main source of aquaculture production, contributing to around 40% (398500 tons per year) of the average yield of aquaculture production in Thailand (range 229700–632200 tons). The larger majority of 95% (380000 tons per year) was from coastal aquaculture, and 5% (18,400 tons per year) was from freshwater aquaculture. About 38% (377100 tons per year, range 523000–193200 tons) of the average yield of aquaculture production was fish (96% from freshwater aquaculture and 4% from coastal aquaculture). Nearly 22% (223500 tons per year, range 66400–382900 tons) were mollusks.

The mean annual value of aquaculture production was estimated at US$2200 million (1 million tons), of which 78% (0.6 million tons) came from coastal aquaculture and the remaining 22% (0.4 million tons) came from freshwater aquaculture [26–46]. The prices of some species slightly increased over the period. For instance, the price of Nile tilapia steadily increased from 20 baht/kg in 1995 to 54 baht/kg in 2015. Likewise, the prices of walking catfish and common silver perch rose by 92% (from 26 to 50) and 88% (from 24 to 45), respectively.

Table 1. The annual yield of aquaculture production and percentage contribution of inland and coastal production in Thailand from 1995 to 2015.

| Year | Coastal Aquaculture (Tons) | | | | | Inland Aquaculture (Tons) | | | | Total Aquaculture Production (Tons) | % of Coastal Aquaculture | % of Inland Aquaculture |
|---|---|---|---|---|---|---|---|---|---|---|---|---|
| | Fish | Shrimps | Mollusks | Others | Total | Fish | Shrimps | Others | Total | | | |
| 1995 | 5132 | 259,540 | 92,835 | 45 | 357,552 | 188,079 | 7792 | 185 | 196,056 | 553,608 | 65 | 35 |
| 1996 | 6235 | 239,500 | 80,183 | 132 | 326,050 | 222,511 | 5586 | 557 | 228,654 | 554,704 | 59 | 41 |
| 1997 | 5652 | 227,560 | 66,408 | 115 | 299,735 | 197,170 | 2159 | 848 | 200,177 | 499,912 | 60 | 40 |
| 1998 | 8794 | 252,731 | 106,128 | 19 | 367,672 | 220,703 | 4764 | 1456 | 226,923 | 594,595 | 62 | 38 |
| 1999 | 7377 | 275,542 | 158,238 | 9 | 441,166 | 242,766 | 8494 | 1352 | 252,612 | 693,778 | 64 | 36 |
| 2000 | 9229 | 309,862 | 147,972 | 9 | 467,072 | 259,695 | 9917 | 1400 | 271,012 | 738,084 | 63 | 37 |
| 2001 | 9588 | 280,007 | 244,949 | 5 | 534,549 | 262,816 | 13,311 | 3569 | 279,696 | 814,245 | 66 | 34 |
| 2002 | 12,251 | 264,923 | 382,918 | 10 | 660,102 | 275,130 | 15,393 | 3978 | 294,501 | 954,603 | 69 | 31 |
| 2003 | 14,599 | 330,725 | 357,944 | 10 | 703,278 | 328,984 | 28,151 | 3990 | 361,125 | 1,064,403 | 66 | 34 |
| 2004 | 17,202 | 360,289 | 358,758 | 22 | 736,271 | 486,382 | 32,583 | 4744 | 523,709 | 1,259,980 | 58 | 42 |
| 2005 | 16,836 | 401,250 | 346,636 | 15 | 764,737 | 506,315 | 28,740 | 4419 | 539,474 | 1,304,211 | 59 | 41 |
| 2006 | 18,346 | 494,401 | 314,116 | - | 826,863 | 498,378 | 25,353 | 3683 | 527,414 | 1,354,277 | 61 | 39 |
| 2007 | 15,523 | 523,226 | 306,571 | 11 | 845,331 | 489,086 | 32,148 | 3861 | 525,095 | 1,370,426 | 62 | 38 |
| 2008 | 16,004 | 506,602 | 285,739 | 23 | 808,368 | 485,060 | 33,189 | 4214 | 522,463 | 1,330,831 | 61 | 39 |
| 2009 | 17,851 | 575,098 | 301,789 | 41 | 894,779 | 490,093 | 26,785 | 5002 | 521,880 | 1,416,659 | 63 | 37 |
| 2010 | 20,205 | 559,644 | 175,615 | - | 755,464 | 469,576 | 22,350 | 4673 | 496,599 | 1,252,063 | 60 | 40 |
| 2011 | 19,126 | 611,194 | 186,730 | - | 817,050 | 358,823 | 21,080 | 4450 | 384,353 | 1,201,403 | 68 | 32 |
| 2012 | 22,330 | 609,552 | 185,861 | - | 817,743 | 431,114 | 18,702 | 4438 | 454,254 | 1,271,997 | 64 | 36 |
| 2013 | 19,256 | 325,395 | 216,835 | - | 561,486 | 413,536 | 18,168 | 4061 | 435,765 | 997,251 | 56 | 44 |
| 2014 | 19,162 | 279,907 | 183,569 | - | 482,638 | 394,915 | 16,906 | 3303 | 415,124 | 897,762 | 54 | 46 |
| 2015 | 19,548 | 294,740 | 194,405 | - | 508,693 | 399,309 | 16,236 | 4300 | 419,845 | 928,538 | 55 | 45 |

Source: DoF [26–46].

## 3.2. Diversity of Species Produced

At least 18 aquatic families were being farmed based on the DoF database (Appendix A; Table A1). The Penaeidae family, specifically whiteleg shrimp (*Penaeus vannamei*) and giant tiger prawn (*Penaeus monodon*), was the largest contributor (38%) to the national aquaculture production, followed by the Mytilidae family (green mussel, *Perna viridis*, and horse mussel, *Musculus senhousia)* at 15% and the Cichlidae family (Nile tilapia, *Oreochromis niloticus,* and java tilapia, *Oreochromis mossambicus*) at 15%.

The relative abundance of the produced species is illustrated in Figure 1. Only the annual yields of the top five species are shown, as these account for about 77% of the total production. Relatively, giant tiger prawns were the most abundant species from 1995 to 2001, followed by green mussels from 2002 to 2004. Since then, whiteleg shrimps have been the most abundant species.

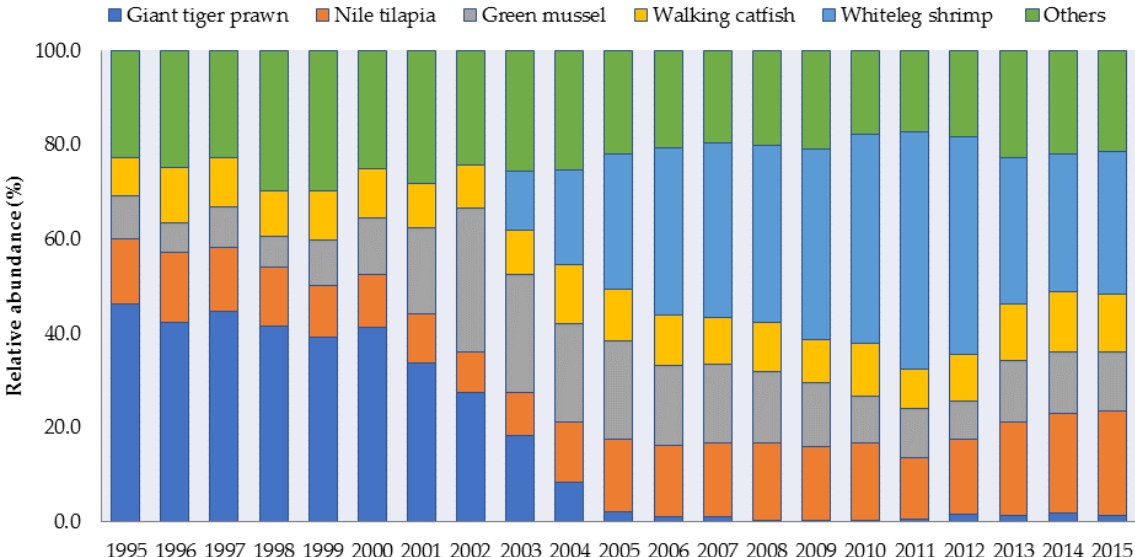

**Figure 1.** The relative abundance of aquatic species that were produced in Thailand from 1995 to 2015. Bar charts show the relative abundance of five major species (i.e., giant tiger prawn, Nile tilapia, green mussel, walking catfish, and whiteleg shrimp) compared to the total weight of all different species present in Thailand's aquaculture production. Based on the DoF [26–46].

In freshwater aquaculture production, fish yield was by far the most significant contributor (94%), followed by giant freshwater prawn and others (6%). Out of all the freshwater produced species, the Nile tilapia (*Oreochromis niloticus*) was the largest contributor (38%), followed by the walking catfish (*Clarias* spp.) (27%), the common silver carp (*Barbonymus gonionotus*) (11%), and others (23%). In coastal aquaculture production, shrimp yields were always the largest contributor (62%), while mollusks and fish accounted for 36% and 2%, respectively. Major cultured species were the whiteleg shrimp (*Penaeus vannamei*) and giant tiger prawn (*Penaeus monodon*). National shrimp culture production was estimated at 260000 tons in 1995 and reached more than 290000 tons in 2015 [26–46]. Thailand's coastal aquaculture faced a significant decline in the production of farmed shrimp from about 600000 tons in 2012 to 325000 tons in 2013 due to disease outbreaks [47]. In several countries, shrimp farming has been promoted to provide economic benefits [13]. The total land area of shrimp farms in Thailand was estimated to expand beyond 74900 ha in 1995 and to peak at 82000 ha in 2003. Then, the 2004–2015 period witnessed a steady decline in shrimp culture. The DoF [46] estimated that the land area of shrimp farms in 2015 shrunk to around 48000 ha. Likewise, shrimp production (specifically the giant tiger shrimp, *Penaeus monodon*) followed a similar trend to the shrimp farmland area. Yields rose from 255900 to 260000 tons from 1995 to 2002, followed by a dramatic decline from 194900 tons in 2003 to 12000 tons in 2015. The decline of giant tiger shrimp yield was, however, mostly due to infectious diseases (e.g., monodon baculovirus, yellow-head virus and white-spot syndrome virus) [48,49].

Farmed mollusk production increased from 3500 to 6000 farms between 1995 and 2015 [26–46]. The dominant species cultivated included the green mussel (*Perna viridis*), the blood cockle (*Anadara* spp.), and oysters (*Saccostrea cucullata, Crassostrea belcheri,* and *Crassostrea iredalei*) [50]. Over 16000 ha of land along the coasts on the Gulf of Thailand and the Andaman Sea were used to support shellfish culture in 2015 [51]. Mollusks are generally farmed along coastlines where wild or hatchery-reared seeds are grown on the seabed bottom or in suspended nets, ropes, wood, or other structures [15]. In 2015, approximately 20% (39600 tons) of total cultured shellfish harvest by weight, worth about US$5.8 million, was gathered from deep-water pound nets and shallow-water pound nets in the coastal waters of Thailand [51].

## 4. The Effects of Aquaculture on the Environment

### 4.1. Land Cover Change

Based on official reports of the DoF, from 1995 to 2015, there was an annual average of 430200 aquaculture farms, with 90% being freshwater farms and 10% being coastal farms. An estimated 199470 ha of land was used for aquaculture farming. Out of the total area, 61% was used for freshwater farms and 39% was used for coastal farms.

Over the twenty-year period (1995–2015), the number of freshwater aquaculture farms dramatically increased from around 131000 farms to more than 540000 farms [26–46]. In 1995, freshwater aquaculture production covered an area of approximately 58000 ha, and it increased to around 128000 ha in 2015. Meanwhile, coastal aquaculture production gradually rose from 32770 farms to 37790 farms between 1995 and 2015. The annual average number of coastal aquaculture farms was about 40884 farms. In total, around 27285 farms (67%) of the annual average of coastal aquaculture farms were potentially for shrimp farming, more than 8200 farms (20%) were for fish farming, and roughly 5300 farms (13%) were for bivalves. Though, on average, shrimp farms made up the majority of coastal farms, their numbers have steadily decreased from 26145 farms to 21082 farms over twenty years. On the other hand, the number of fish and shellfish farms have risen from 3082 to 10696 farms and 3541 to 6015 farms, respectively. Figure 2 illustrates the changes in the area that were used for coastal aquaculture production in 25 Thai provinces during 1995–2015. The Surat Thani province was the most important area that was used for coastal aquaculture farms and accounted for about 11% of the total land that was used for coastal aquaculture production. The available data for each province are shown in Appendix A; Table A2.

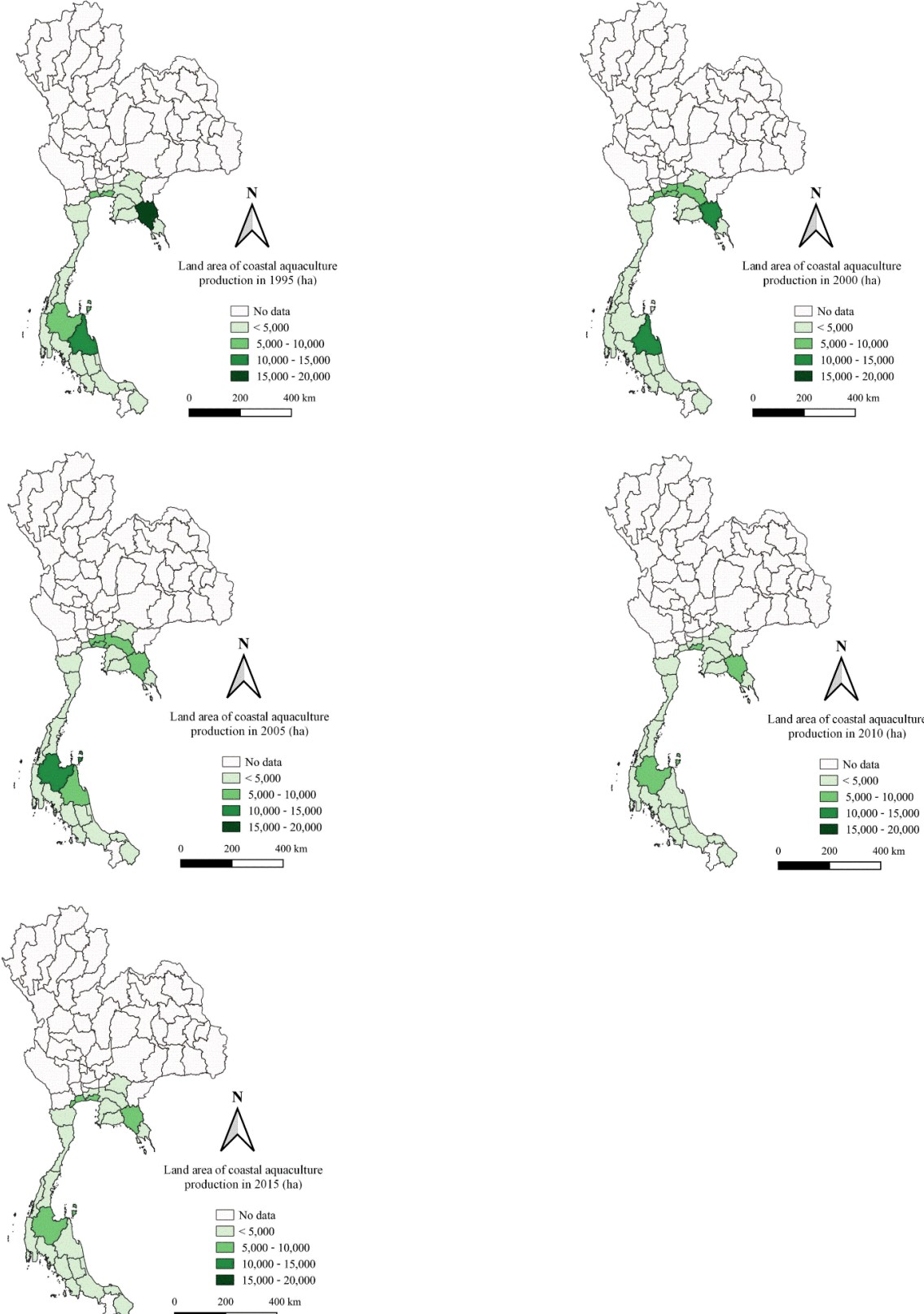

**Figure 2.** Changes in land used for coastal aquaculture production in 25 Thai provinces from 1995 to 2015. Based on the DoF [26–46].

Figure 3 shows the average yield per hectare of all species present in Thailand's coastal aquaculture and the land area used for production. From 1995 to 2015, the yield ranged from 4 to 13 tons/ha, with an average of 8.0 ± 3.0 tons/ha. The land area of coastal aquaculture production surged and then followed a downward trend. It is estimated that the total land area of coastal aquaculture production in Thailand grew to around 79200 ha in 1995 and reached a peak of around 95000 ha in 2003, the highest number over the last two decades [26–46]. After that, there was an abrupt decrease of 12% between 2003 and 2004 as a result of disease outbreaks in shrimp [48,52] and the 2004 tsunami [53]. After that, the land area steadily decreased from 2004 to 2015. The DoF [46] suggested that land area of coastal production in 2015 shrunk to around 65800 ha.

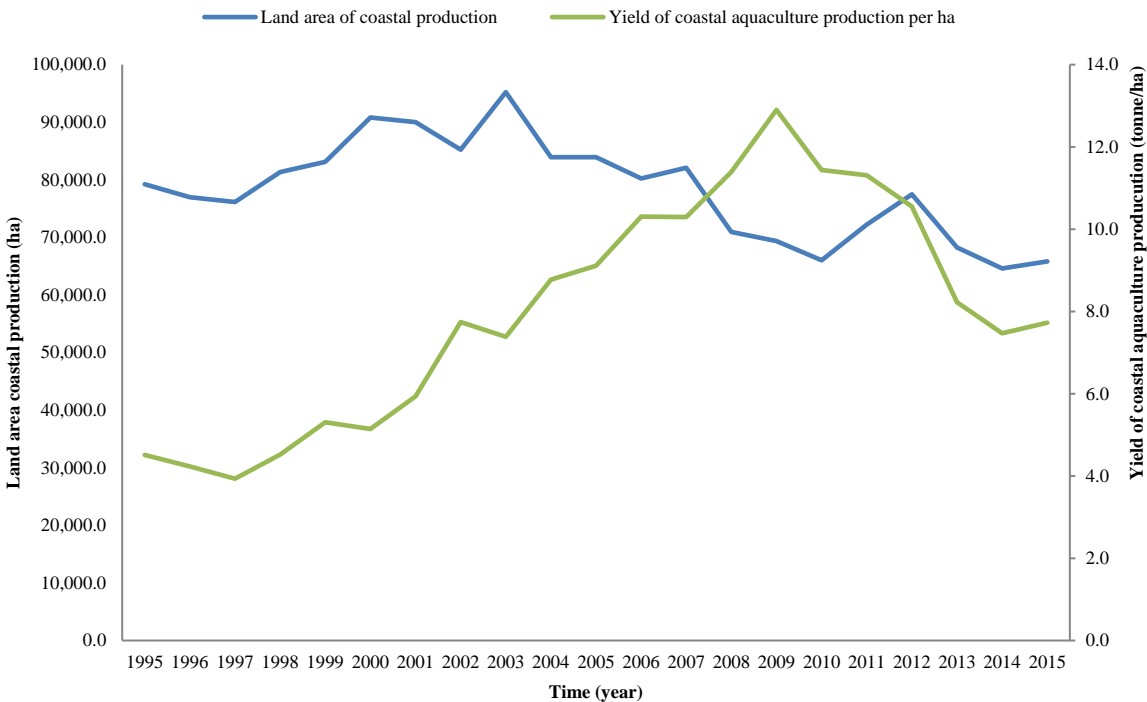

**Figure 3.** Total yield per hectare (green) of all species present in coastal aquaculture and the land area used for coastal production (blue). Based on the DoF [26–46].

### 4.2. Degradation of Mangrove Forest

Apparently, the increase of shrimp culture production has degraded and deforested coastal areas, including mangrove forests [13,54,55]. Several studies have suggested that the mangrove area has a significant role to play in the provision of human food, nursery habitats for marine animals, coastal protection, flood control, sediment trapping, and water treatment [13,15].

Thailand's mangrove area dramatically decreased between 1961 and 1996, from 367000 ha to 167582 ha (Table 2). After a period of short increase, Thailand's mangrove forest area again steadily decreased from 252765 ha in 2000 to 245534 ha in 2014 [56,57]. It is estimated that Thailand lost about 122,300 ha of mangroves over a half-century from 1961 to 2014 (33% of the area in 1961) [56,57]. Menasveta [55] indicated that approximately 65000 ha of mangroves were converted to shrimp ponds from 1961 to 1996, making this the main cause of mangrove deforestation in Thailand. However, since the late 1990s, concerns have been raised about the sustainability of these intensive practices. Consequently, Thailand has formulated and modified its policies and plans to restore and rehabilitate mangrove forests across the country [16,54]. For example, the Fisheries Act prohibits pond construction in public mangrove areas [58] because shrimp farms are not opened in mangrove areas [59]. As a result of increasing awareness in the country, the annual rate of mangrove area lost has gradually decreased in recent years.

**Table 2.** Estimated total mangrove forest area in Thailand between 1961 and 2014.

| Year | Estimated Total Mangrove Forest Area (ha) | Mangrove Area Changes | |
| --- | --- | --- | --- |
| | | ha | % |
| 1961 | 367,900 | | |
| 1975 | 312,700 | −55,200 | −15 |
| 1979 | 287,308 | −25,392 | −8 |
| 1986 | 196,436 | −90,872 | −32 |
| 1989 | 180,559 | −15,877 | −8 |
| 1991 | 173,821 | −6,738 | −4 |
| 1993 | 168,683 | −5,138 | −3 |
| 1996 | 167,582 | −1,100 | −1 |
| 2000 | 252,765 | 85,183 | 51 |
| 2004 | 233,308 | −19,457 | −8 |
| 2009 | 244,010 | 10,702 | 5 |
| 2014 | 245,534 | 1,524 | 1 |

Source: Adapted from Menasveta [55] and DMCR [56].

### 4.3. Impact of Exotic Species

Against a backdrop of stagnating aquaculture production, Thailand's shrimp production switched from farming tiger shrimp to whiteleg shrimp. This species is originally native to the eastern Pacific coast from Sonora, Mexico in the north, through Central and South America as far south as Tumbes in Peru [60]. This species was introduced to the Thai aquaculture in 2000 [61] as a disease-resistant species [52]. As a result, whiteleg shrimp production has rapidly burgeoned from around 132000 tons to more than 281000 tons from 2003 to 2015 [26–46].

Exotic species are a threat to global biodiversity [62,63]. Exotic species commonly contribute to the decline and extinction of native species, but some others can contribute economic or social benefits to recipient communities [63–65]. According to the DoF database, many exotic species in Thailand (e.g., *Penaeus vannamei*, *Oreochromis niloticus*, and *Barbonymus gonionotus*) are major species in aquaculture. There are about 40 exotic species recorded in Thai aquaculture farms [61], with seven species (*Clarias gariepinus*, *Hypostomus* spp., *Pterygoplichthys* sp., *Arapaima gigas*, *Serrasalmus* spp., *Pomacea gigas* and *Pomacea canaliculate*) considered as invasive. Meanwhile, two species (*Trachinotus blochii* and *Artemia* spp.) have a beneficial effect on aquaculture production. Several exotic species are a major threat to marine or freshwater ecosystems, e.g., the Amazon apple snail (*Pomacea canaliculata*) [66]. This species was initially introduced from South America to Southeast Asia in the 1980s as a local food resource and as a potential gourmet export item [65]. It rapidly escaped or was released in agricultural areas, lakes, watercourses, and wetlands. It became a serious pest in rice paddies in many Southeast Asian countries, including Thailand [65,67], and is part of the 100 of the world's worst invasive species [67]. In recent years, international agreements such as the Sustainable Development Goals (SDG 15) and the Convention on Biological Diversity (Aichi Biodiversity Target 9) have prioritized the control and/or eradication of alien species and the minimization of their impact on land and water ecosystems.

### 4.4. Water Pollution

Eutrophication, a process that is caused by the excessive input of nutrients (e.g., phosphorus and nitrogen), is widely recognized as a severe threat to the environment [15,68]. It negatively affects water quality and eventually leads to ecological damage [68]. The intensification of aquaculture production is a major source of eutrophication [15,52], mainly due to the release of untreated wastewater and sewage sludge from fish and shrimp farms [69,70]. The water quality of overstocked and/or overfed fish farms is commonly poor as a result of the decomposition of waste feed and fish feces, and its discharge can have negative effects on surrounding water sources [69]. Effluents from such farms unload a massive amount of nutrients into coastal and estuarine waters, often stimulating the rapid growth of primary producers in water ecosystems, such as algae and plankton [68]. Cheevaporn and

Menasveta [71] documented that the blue-green algae (*Trichodesmium erythraem* and *Notilluca* sp.) bloomed in the Gulf of Thailand due to the disposal of untreated sewage. Luo, et al. [72] indicated that the continuous accumulation of certain compounds, e.g., nitrogen, can lead to acidification and cause adverse effects on aquatic plants and animals, with significant biotic damage.

The problems of effluent discharge from aquaculture farms have been widely discussed [15]. During recent decades, authors have examined techniques for environmentally friendly aquaculture to reduce the inputs of nitrogen and phosphorus from point-source effluents to water bodies [73–75]. Biofloc technology has been gaining popularity as an efficient alternative water management system [73,75,76]. It combines the removal of nutrients from water with the production of microbial biomass, which can be used by the culture species in situ as feed supplements [77]. Furthermore, the concept and practice of integrated multi-trophic aquaculture constitutes one way of reducing water pollution problems that are caused by aquaculture activity [78]. Multi-trophic aquaculture is based on the concept that waste from one species, such as uneaten feed, feces, and metabolic excretion, is useful for the growth of other species, thus forming a natural self-cleansing mechanism [79]. Many countries, e.g., the Philippines, Malaysia, Vietnam, China, and Thailand, have incorporated this practice by culturing fish species in combination with seaweed to increase economic benefits and reduce negative environmental impacts from aquaculture activities [80].

## 5. Prospects for Sustainable Aquaculture in Thailand

In the face of population growth, increasing demand for animal protein, and the limitation of expanding wild fishery harvests, aquaculture production presents an opportunity to increase seafood production [81,82]. Thai aquaculture production has rapidly developed during the last few decades and has been responsible for most of the yield increase of fish supply. The promotion of aquaculture production has become one of the key strategies in Thailand and is considered key to provide food security as well as developing national economic activities (Office of the National Economic and Social Development (NESDB, 2019). According to the current five-year National Economic and Social Development Plan (2017–2021), the Government of Thailand announced its policy that encourages the country's aquaculture production. The DoF is the main implementing agency in the fisheries and aquaculture sector under the administrative control of the Ministry of Agriculture and Cooperatives. However, this industry has come under scrutiny with concerns regarding environmental degradation [76]. Governmental agencies have made several attempts to improve and promote a sustainable farming industry through the reformation of Thai aquaculture, e.g., the Agricultural Standards Act B.E. 2551 (2008), the Thai Agricultural Standard on Good Aquaculture Practices for marine shrimp (TAS 7401-2014), and the shrimp Code of Conduct. Governmental agencies are supporting the development of new aquaculture technologies and tools and have been disseminating them to farmers to support sustainable aquaculture practices [22].

Increasingly, attention is being given to shrimp farming in Thailand due to suitable geographical conditions and recent technologies that have boosted its productivity [23]. As a result, the total land area for shrimp aquaculture has rapidly expanded in the last few decades. Gentry, et al. [82] and Sorgeloos [83] argued that coastal areas in many countries that are suitable for marine aquaculture could meet foreseeable seafood demands, specifically mollusk production. Though farming seafood in the ocean can have potential for the future growth of aquaculture production, environmentally sensitive or high biodiversity areas, such as coral reefs, should be protected from farming industries [82]. The development of ports and harbours for accessing seafood markets and farming infrastructures need to take into account the growth of future mariculture [82].

Though aquaculture production systems could contribute to provide food and nutrition for people, as well as to develop the national economy, an unsustainable expansion of the industry poses a significant threat to ocean resources, coastal resources, and the global environment. A growing issue is the massive amount of wild fish, particularly trash fish, that are needed to feed in the farmed fish and shellfish industries [84,85]. Several studies have investigated alternative sources of protein (e.g., algae

meal, wheat gluten, corn gluten, and insects) to replace and reduce the use of fishmeal and fish oil in aquafeed production [86,87].

Furthermore, although water quality and quantity are of paramount importance for aquaculture production, it appears that proper water resource management for sustainable aquaculture has remained a major challenge in Thailand [23]. To tackle this issue, low- and high-tech farming practices that are designed for eco-friendly aquaculture, such as the integration of cultures from different trophic levels, the integration of rice-fish farming, and the integration of production systems with livestock and agriculture, can be proper solutions [78,79,88]. Equally important are innovative technologies such as microbial management of farming systems that can offer a balanced solution between environmental remediation, economic benefits, and social acceptability [73,75,76]. In some cases, extensive (low-tech) aquaculture can be the most sustainable option, where reduced food production can be compensated for by other ecosystem services of aquaculture ponds [89]. Interestingly, the new practice of intensive shrimp farming in Thailand is a good example of a sustainable aquaculture practice. This practice implements a zero-water-exchange system by recirculating wastewater from shrimp ponds into ponds that are stocked with tilapia or *Caulerpa* seaweed. These so-called "shrimp toilets" aid in waste removal and significantly improve the sustainability of shrimp farming [90]. It is interesting to see that these solutions do not need necessarily require high-technology, and they are often economically profitable as well. Thus, future policies and research must focus on developing easy-to-adopt sustainable aquaculture practices and disseminating such information and technology to farmers.

Finally, natural disasters, such as tsunamis, flooding, and animal disease outbreaks can have destructive effects on aquaculture production [13]. For example, in the past few years, Thailand's shrimp aquaculture production has been disrupted by disease outbreaks, such as the early mortality syndrome/acute hepatopancreatic necrosis disease (EMS/AHPND) [47]. As Thailand is still facing the risk of aquatic animal diseases in aquaculture, the Government of Thailand has invested in research at universities and quasi-public institutions such as the Thai National Center for Genetic Engineering and Biotechnology (BIOTEC) to address this problem [59].

## 6. Conclusions

In this paper, we reviewed the evolution of aquaculture production in Thailand under a perspective of environmental sustainability. We drew several important conclusions. Firstly, Thailand's aquaculture production has rapidly developed during the last few decades and has been responsible for an increase in seafood supply. However, despite its substantial economic growth, this rapid development has led to numerous environmental problems, e.g., the loss of ecologically sensitive land as a result of land use for aquaculture production, the introduction of exotic species for production purposes leading to damages of ecosystem compositions, and eutrophication due to the discharges of aquaculture farms. Hence, the development and implementation of effective management approaches are urgently needed. From this perspective, several novel approaches to facilitate responsible aquaculture practices have been proposed, and these involve both traditional and advanced technology, e.g., the integration of aquaculture production systems with livestock and agriculture, the development of alternative sources of protein to replace and reduce the use of fishmeal in aquaculture feed, water quality treatment, and the microbial management of farming systems. These practices can be the basis for viable long-term solutions for sustainable aquaculture production and environmental practices in the future.

**Author Contributions:** Conceptualization, T.S., and P.G.; The structure of the manuscript and analysis, T.S., L.H., C.L., N.S., P.S., and P.G.; Writing—original draft, T.S.; Writing—review and editing, T.S., L.H., C.L., N.S., P.S., and P.G. All authors have read and agreed to the published version of the manuscript.

**Funding:** Thaksin University supported this work through a Ph.D. scholarship.

**Acknowledgments:** Special thanks are extended to anonymous reviewers and numerous colleagues for an informal review of our manuscript. We thank Srisuwan Kuankachorn, Roschong Boonyarittichaikij, Chananchida Sang-aram, and Wisarut Junprung for their helpful comments and suggestions. We also thank Thailand's Department of Fisheries for providing the fisheries database.

**Conflicts of Interest:** The authors declare no conflict of interest. The sponsor had no role in the design of the study; in the collection, analyses, or interpretation of data; in the writing of the manuscript, and in the decision to publish the results.

## Appendix A

**Table A1.** Taxonomic composition and group of species produced in aquaculture production in Thailand from 1995 to 2015.

| Taxon | Yields of Aquaculture Production (Tons) | | | | | Average Taxonomic Composition of Aquaculture Production (Tons) | Percentage of Taxonomic Composition of Aquaculture Production |
|---|---|---|---|---|---|---|---|
| | 1995 | 2000 | 2005 | 2010 | 2015 | | |
| **Fish** | | | | | | | |
| **Mytilidae** | **55,395** | **88,759** | **270,677** | **123,879** | **115,543** | **149,521 ± 83,179** | **14.9** |
| *Perna viridis* | 51,184 | 88,759 | 270,677 | 123,879 | 115,543 | 149,062 ± 8378 | 14.9 |
| *Musculus senhousia* | 4211 | - | - | - | - | 878 ± 1971 | <0.1 |
| **Cichlidae** | **76,057** | **82,391** | **203,896** | **204,726** | **205,974** | **147,807 ± 61,864** | **14.7** |
| *Oreochromis niloticus* | 76,054 | 82,363 | 203,737 | 204,680 | 205,896 | 147,729 ± 61,852 | 14.7 |
| *Oreochromis mossambicus* | 3 | 28 | 159 | 46 | 78 | 77 ± 62 | <0.1 |
| **Clariidae** | **44,120** | **76,000** | **142,205** | **140,763** | **114,179** | **104,625 ± 34,816** | **10.4** |
| *Clarias* spp. | 44,120 | 76,000 | 142,205 | 140,763 | 114,179 | 104,625 ± 34,816 | 10.4 |
| **Arcidae** | **14,403** | **45,657** | **56,853** | **40,979** | **58,991** | **54,892 ± 20,813** | **5.5** |
| *Anadara* spp. | 14,403 | 45,657 | 56,853 | 40,979 | 58,991 | 54,892 ± 20,813 | 5.5 |
| **Cyprinidae** | **33,599** | **54,482** | **70,361** | **46,490** | **33,461** | **49,783 ± 13,155** | **5.0** |
| *Cyprinus* spp. | 3556 | 5539 | 5036 | 2417 | 1285 | 4341 ± 2394 | 0.4 |
| *Barbonymus gonionotus* | 27,432 | 46,276 | 60,643 | 42,049 | 30,498 | 42,442 ± 11,441 | 4.2 |
| Chinese major carps mixed species | 653 | 438 | 285 | 354 | 200 | 363 ± 226 | <0.1 |
| *Labeo rohita* | 1480 | 1172 | 3196 | 1169 | 1101 | 1848 ± 1108 | 0.2 |
| *Cirrhinus mrigala* | 478 | 1052 | 1201 | 501 | 377 | 790 ± 343 | 0.1 |
| **Osphronemidae** | **17,321** | **23,233** | **41,377** | **38,957** | **18,621** | **29,125 ± 9,231** | **2.9** |
| *Trichopodus pectoralis* | 16,714 | 21,577 | 35,867 | 34,419 | 14,956 | 26,142 ± 8,053 | 2.6 |
| *Trichopodus* spp. | 259 | 169 | 58 | 5 | 4 | 112 ± 143 | <0.1 |
| *Osphronemus goramy* | 348 | 1487 | 5452 | 4533 | 3661 | 2827 ± 1591 | 0.3 |
| **Pangasiidae** | **7308** | **13,231** | **27,252** | **27,455** | **19,790** | **19,488 ± 7387** | **1.9** |
| *Pangasianodon hypophthalmus* | 7308 | 13,226 | 26,446 | 27,027 | 19,790 | 19,157 ± 7173 | 1.9 |
| *Pangasius larnaudii* | - | 5 | 806 | 428 | 19,060 | 730 ± 33 | <0.1 |
| **Ostreidae** | **23,037** | **13,556** | **19,106** | **10,757** | **19,871** | **19,121 ± 5,885** | **1.9** |
| *Saccostrea cucullata* | 23,037 | 13,556 | 19,106 | 10,757 | 19,871 | 19,121 ± 5885 | 1.9 |
| **Latidae** | **3882** | **7752** | **14,219** | **17,415** | **17,250** | **11,936 ± 4939** | **1.2** |
| *Lates calcarifer* | 3882 | 7752 | 14,219 | 17,415 | 17,250 | 11,936 ± 4939 | 1.2 |
| **Channidae** | **6430** | **4527** | **12,507** | **4639** | **3641** | **6330 ± 2453** | **0.6** |
| *Channa striata* | 5791 | 4447 | 12,300 | 4340 | 3075 | 5968 ± 2484 | 0.6 |
| *Channa micropeltes* | 639 | 80 | 207 | 299 | 566 | 362 ± 295 | <0.1 |
| **Serranidae** | **674** | **1332** | **2582** | **2790** | **2258** | **2139 ± 906** | **0.2** |
| *Epinephelus* spp. | 674 | 1332 | 2582 | 2790 | 2258 | 2139 ± 906 | 0.2 |
| **Anabantidae** | **949** | **470** | **2965** | **486** | **223** | **871 ± 773** | **0.1** |
| *Anabas testudineus* | 949 | 470 | 2965 | 486 | 223 | 871 ± 773 | 0.1 |
| **Eleotridae** | **67** | **5** | **98** | **114** | **78** | **70 ± 46** | **<0.1** |
| *Oxyeleotris marmorata* | 67 | 5 | 98 | 114 | 78 | 70 ± 46 | <0.1 |
| **Synbranchidae** | **1** | **38** | **65** | **-** | **-** | **44 ± 115** | **<0.1** |
| *Monopterus albus* | 1 | 38 | 65 | - | - | 44 ± 115 | <0.1 |
| **Notopteridae** | **49** | **5** | **28** | **1** | **4** | **12 ± 19** | **<0.1** |
| *Notopterus* spp. | 49 | 5 | 28 | 1 | 4 | 12 ± 19 | <0.1 |
| **Mugilidae** | **-** | **-** | **28** | **-** | **-** | **2 ± 6** | **<0.1** |
| *Mullet group* | - | - | 28 | - | - | 2 ± 6 | <0.1 |

**Table A1.** *Cont.*

| Taxon | Yields of Aquaculture Production (Tons) | | | | | Average Taxonomic Composition of Aquaculture Production (Tons) | Percentage of Taxonomic Composition of Aquaculture Production |
|---|---|---|---|---|---|---|---|
| | 1995 | 2000 | 2005 | 2010 | 2015 | | |
| Fish mixed group | 2754 | 5458 | 5568 | 5945 | 3378 | 4944 ± 1508 | 0.5 |
| Shrimp | | | | | | | |
| Penaeidae | 258,398 | 309,206 | 401,150 | 559,427 | 294,703 | 379,601 ± 134,883 | 37.9 |
| *Penaeus merguiensis* | 1813 | 3562 | 508 | 318 | 237 | 1391 ± 1328 | 0.1 |
| *Penaeus monodon* | 255,890 | 304,988 | 26,055 | 5105 | 12,098 | 119,625 ± 120,971 | 11.9 |
| *Litopenaeus vannamei* | - | - | 374,487 | 553,899 | 281,918 | 258,143 ± 239,846 | 25.7 |
| *Metapenaeus* spp. | 695 | 656 | 100 | 105 | 450 | 442 ± 421 | <0.1 |
| Palaemonidae | 7792 | 9917 | 28,740 | 22,350 | 16,236 | 18,467 ± 9723 | 1.8 |
| *Macrobrachium rosenbergii* | 7792 | 9917 | 28,740 | 22,350 | 16,236 | 18,467 ± 9723 | 1.8 |
| Shrimp mixed group | 1142 | 656 | 100 | 217 | 37 | 480 ± 473 | <0.1 |
| Carb | | | | | | | |
| Carb mixed group | 45 | 9 | 15 | - | - | 22 ± 36 | <0.1 |
| Other aquatic animals | 185 | 1400 | 4419 | 4673 | 4300 | 3261 ± 1559 | 0.3 |
| Total aquaculture production | 553,608 | 738,084 | 1,304,211 | 1,252,063 | 928,538 | 1,002,540 ± 309,395 | 100 |

Source: Adapted from the DoF [26–46]. All species that are mentioned on the list of landings of Thailand's marine fisheries were identified by using a guide to marine fishes in Thailand, the international online fish database (http://www.fishbase.org), and the IUCN red list of threatened species (http://www.iucnredlist.org/about).

**Table A2.** The amount of area used for coastal aquaculture production in 25 Thai provinces from 1995 to 2015. Source: Based on the DoF [26–46].

| Province | Year (ha) | | | | | Average |
|---|---|---|---|---|---|---|
| | 1995 | 2000 | 2005 | 2010 | 2015 | |
| Trat | 2621 | 1586 | 3472 | 2238 | 1548 | 2075 ± 608 |
| Chanthaburi | 16253 | 13802 | 6175 | 6361 | 5824 | 7995 ± 2989 |
| Rayong | 1386 | 663 | 1269 | 1575 | 1088 | 1244 ± 349 |
| Chin Buri | 1553 | 1778 | 635 | 1730 | 540 | 1265 ± 615 |
| hachengsao | 2850 | 9096 | 8659 | 4101 | 3612 | 6295 ± 2526 |
| Prahin Buri | 362 | 1248 | 1923 | 1888 | 985 | 1351 ± 659 |
| Samut Prakan | 5879 | 8499 | 7327 | 5112 | 7773 | 7193 ± 1494 |
| Bangkok | 2811 | 2970 | 2957 | 875 | 3757 | 2626 ± 868 |
| Samut Sakhon | 6273 | 5998 | 2979 | 4833 | 6774 | 5140 ± 1357 |
| Samut Songkhram | 4387 | 5878 | 3948 | 4539 | 5577 | 5358 ± 836 |
| Phetchaburi | 1809 | 2102 | 1286 | 2724 | 4728 | 2563 ± 1450 |
| Prachuap Khiri Khan | 1004 | 1967 | 1644 | 2348 | 1049 | 1977 ± 975 |
| Chumphon | 2248 | 1296 | 2946 | 1792 | 944 | 1870 ± 530 |
| Surat Thani | 8378 | 4814 | 12887 | 7890 | 6496 | 8749 ± 2650 |
| Nakhon Si Thanmmarat | 10734 | 11317 | 8233 | 3848 | 2493 | 7215 ± 3342 |
| Songkhla | 2975 | 2281 | 4265 | 1841 | 1220 | 2483 ± 989 |
| Phatthalung | 249 | 566 | 1097 | 190 | 90 | 361 ± 282 |
| Pattani | 892 | 819 | 1571 | 920 | 244 | 864 ± 426 |
| Narathiwat | 13 | 20 | 82 | 54 | 27 | 36 ± 18 |
| Ranong | 959 | 666 | 1204 | 929 | 656 | 871 ± 257 |
| Phangnga | 1295 | 1158 | 1604 | 1173 | 729 | 1339 ± 389 |
| Phuket | 340 | 248 | 316 | 266 | 129 | 288 ± 79 |
| Krabi | 1141 | 969 | 1040 | 1351 | 603 | 1152 ± 288 |
| Trang | 1208 | 1877 | 1492 | 1514 | 772 | 1345 ± 366 |
| Satun | 1602 | 1281 | 1872 | 1480 | 1350 | 1572 ± 299 |

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
