# Peer review of "Aquaculture Production and Its Environmental Sustainability in Thailand: Challenges and Potential Solutions"

_sustainability, doi:10.3390/su12052010_

Round 1

Reviewer 1 Report

Interesting and valuable work, bringing new information to learning, which is why I recommend it for printing.
Streszcsena corresponds to the content of the article. What is missing, as in the chapter, is the lack of specific solutions as to what should be done in this area. The conclusions are not only conclusions and a miserable one. You should either complete the summary or replace it with an application and provide 2 or 3 applications resulting from the work.
This is the only weak point of work.
The correct purpose and scope of work. Good graphic design, as well as analysis of the presented tables and drawings.
The authors have comprehensively approached the solution of the assumed goal of work and I believe that the goal was achieved. You can see that the authors are specialists in this field. They described all hazards and conditions of Aquaculture production very well, as well as proposed prospects for sustainable aquaculture in Thailand. I particularly value chapter 5.
To sum up, I conclude that the work is valuable and should be published after previous supplementing or changing chapter 6 (Conclusions).

Reviewer 2 Report

The paper requires minor editing. Otherwise, a good one. 

Author Response

The authors would like to thank the reviewers for your considerations. We revised some parts of the manuscript. Please see the reversion of manuscript. Thank you so much. 

Reviewer 3 Report

The paper has two weaknesses. First, there is no clear methodology in the paper. It is therefore difficult to see how the results were generated. Second, the paper aims to provide insights into environmental impacts of aquaculture and an assessment of sustainability. Reading the paper does not give the reader confidence that impacts are being analyzed and sustainability assessed using reliable scientific methods. There are no clear methods for assessment of the impacts on diversity of species, yield of aquaculture production, degradation of magrove forests, water pollution, and land cover changes that can scientifically be attributed to aquaculture. The point here is attribution, i.e. how are the authors eliminating impacts from other variables? How are they measuring the robustness of their results in the assessment of sustainability?

Round 2

Reviewer 3 Report

The responses to my comments are now satisfactory.